# The impact of classroom interaction on willingness to communicate: The mediating roles of speaking self-efficacy and foreign language enjoyment

Zhiyu Liu[1], Dawei Sun[1], Lei Zhang[1], Xianzhi Wang[2], Yanchao Yang[3]*

1 College of Foreign Languages, North China University of Science and Technology, Tangshan, Hebei, People's Republic of China, 2 Department of International Cooperation, North China University of Science and Technology, Tangshan, Hebei, People's Republic of China, 3 Institute of International Language Services Studies, Macau Millennium College, Macau SAR, People's Republic of China

* yangyanchao@mmc.edu.mo

## Abstract

In the realm of second language acquisition, classroom interaction (CI) is pivotal for learners' language development, and willingness to communicate (WTC) is widely recognized as a key factor in successful language learning. However, previous studies have not fully addressed the impact of CI on WTC, particularly regarding the mediating roles of speaking self-efficacy (SSE) and foreign language enjoyment (FLE). Therefore, this study aims to address these gaps. The research recruited 623 undergraduate students from three universities in China. Data were collected using a self-designed questionnaire that demonstrated strong reliability and validity. Structural equation modeling was employed to test the proposed hypotheses, with bootstrap analysis used to assess the significance of mediation effects. The results revealed that CI had a significant positive direct impact on WTC. Moreover, both SSE and FLE significantly mediated the relationship between CI and WTC. Notably, CI also affected WTC through the chain mediating effect of SSE and FLE. These findings provide empirical support for relevant theories and offer practical implications for language educators, such as fostering interactive classroom environments to boost students' SSE and FLE, ultimately promoting their WTC in language learning. The limitations and directions for future research were also discussed.

## 1 Introduction

In second language acquisition (SLA), classroom interaction plays a crucial role in shaping learners' linguistic development. Given that language learning is inherently social, the classroom serves as a key site where interaction facilitates language practice, enhances fluency, and supports the development of communicative competence [1–3]. Specifically, classroom interaction (CI), which involves the interaction between

**Data availability statement:** All relevant data are within the manuscript and its Supporting Information files.

**Funding:** This research was funded by The Education Department of Hebei Province, China (grant numbers: 2024WYJG016 and 2024YYJG093). The funder had no role in study design, data collection and analysis, decision to publish, or preparation of the manuscript

**Competing interests:** The authors have declared that no competing interests exist.

teachers and students as well as among students in the classroom [4], significantly affects student's academic achievements, perceptions of occupational suitability, and communication intentions [5,6]. Defined as readiness to engage in communication at a given moment with a particular person or group [7], willingness to communicate (WTC) is a critical predictor of successful language learning. Numerous existing studies have acknowledged the significance of WTC, as it promotes language practice, boosts confidence, and improves language proficiency and motivation [8–10].

While previous studies [6,11] have provided valuable insights into the relationship between CI and WTC, they are limited by a reliance on respondents' perception of CI (a cognitive variable), rather than examining the impact of actual classroom interactive behaviors that represent the behavioral dimension of CI. Furthermore, although studies [12,13] have investigated variations in WTC across different classroom contexts (e.g., whole class, group, and dyadic interactions), their findings are based on small sample sizes (8 or 23 participants), which may limit the generalizability of the results. These limitations underscore the need for future research that explores the behavioral aspects of CI using larger and more diverse participant samples to obtain more robust and generalizable conclusions. In addition, studies [14–18] offer both theoretical and empirical foundations for examining whether the impact of CI on WTC is mediated by cognitive factors (e.g., speaking self-efficacy, SSE) and affective factors (e.g., foreign language enjoyment, FLE). Bandura's self-efficacy theory [14] and the control-value theory of achievement emotions [15] suggest that learners' beliefs and emotional experiences play a critical role in language learning. Empirical research [16–18] has also shown that classroom factors such as teacher support and classroom environment influence WTC indirectly through variables like SSE and FLE. These insights collectively point to the relevance of exploring SSE and FLE as potential mediators in the CI-WTC relationship. Drawing on these theoretical and empirical insights, this study proposes the mediation model in which CI impacts WTC both directly and indirectly. The following sections provide a detailed review of the literature on the relationships among CI, WTC, SSE, and FLE, and outline the corresponding research hypotheses.

## 1.1 The relationship between classroom interaction and willingness to communicate

The well-known social interactionist theory (SIT) posits that social interaction plays an essential role in the development of cognitive and language skills. According to Vygotsky and Cole [19], learning occurs when individuals engage in social interactions with more knowledgeable others, such as teachers and peers, who can offer guidance, support, and scaffolding. Building on the SIT, it is proposed that CI helps to enhance students' WTC. As students actively participate in classroom interactional activities, they are provided opportunities to practice their language skills. This engagement not only elevates their language proficiency but also boosts their motivation and confidence for communication.

Existing empirical studies have provided valuable insights into the relationship between CI and WTC. Studies [6,11] have demonstrated that students' perceptions

of classroom interaction, including group interaction and interaction with teachers, affected their WTC in a range of ways. In addition to examining the influence of students' perceptions, prior research [12,13] has explored the variation in WTC across different classroom contexts, such as whole class interaction, group interaction, and dyadic interaction, although their findings were based on small sample sizes (8 or 23 participants). On the basis of SIT and the findings from previous studies, this study aims to explore the influence of CI (a behavioral variable) upon WTC using large-scale quantitative data. Accordingly, the following hypothesis is proposed for further investigation:

Hypothesis 1: Classroom interaction positively predicts willingness to communicate in English, implying that greater classroom interaction fosters an increased willingness to communicate in English in EFL settings.

## 1.2 The relationship between classroom interaction, speaking self-efficacy, and willingness to communicate

Self-efficacy, a concept introduced by Bandura, denotes an individual's belief in their ability to organize and perform the actions required to achieve specific objectives [14]. In educational contexts, self-efficacy is pivotal in shaping students' motivation, levels of engagement, and overall achievement. Specifically, in second language acquisition, high self-efficacy leads to enhanced perseverance in tackling difficult tasks, increased willingness to embrace challenges, and the more frequent use of communication strategies [20,21]. Speaking self-efficacy (SSE) refers to an individual's confidence in their ability to properly perform grammar, usage, communication, and interaction [22]. It is an application of Bandura's broader self-efficacy theory to the context of language learning. Research [17] has demonstrated that language learners with high SSE are more likely to engage in speaking activities, keep persevering when confronted with challenges, and achieve higher speaking proficiency.

The central concept of reciprocal determinism in social cognitive theory (SCT), set forth by Bandura, states that human behavior results from a dynamic and reciprocal interaction among three key factors: person (individual with learned experiences), environment (external social context), and behavior (response to stimuli to achieve goals) [23]. According to reciprocal determinism, personal beliefs (cognitive or emotional) are of fundamental importance in shaping behaviors, while environmental contexts offer the opportunities and stimuli for these behaviors to occur [23]. In the context of this study, it is hypothesized that CI can enhance students' SSE by providing a supportive and stimulating environment through methods such as modelling, feedback, and cooperative activities. This, in turn, boosts students' motivation and confidence in engaging in communication, which is central to their WTC.

While previous studies have explored self-efficacy as a mediator in the relationship between learning environments and students' learning engagement or achievement [24–28], few have examined its role in mediating the relationship between CI and WTC. For instance, the study [26] found that active learning classroom structures enhanced self-efficacy in traditional classrooms but reduced it in technology-enhanced environments, with self-efficacy being a significant predictor of academic achievement. Similarly, the studies [27,28] demonstrated students' self-efficacy also mediates the relationship between learning environment (like learner-learner interaction, learner-instructor interaction, as well as teacher supports) and learning engagement. However, despite its significant mediating role in various learning contexts, self-efficacy is still an under-explored variable in the relationship of CI and WTC, particularly with regard to the mediating role of SSE. Considering this research gap and the theoretical framework of reciprocal determinism in SCT, the current investigation seeks to test the following hypothesis:

Hypothesis 2: Speaking self-efficacy acts as a mediator in the relationship between classroom interaction and willingness to communicate.

## 1.3 The relationship between classroom interaction, foreign language enjoyment, and willingness to communicate

Academic emotions are the feelings of learners that related to their learning process and outcomes, a process that involves both enduring emotional states and complex personal experiences [15]. These emotions can be classified

as either positive or negative. Among them, foreign language enjoyment (FLE) is defined as the positive emotional response students experience while learning or using a foreign language, including the feelings of joy, excitement, and satisfaction [29]. Research has demonstrated that FLE contributes to greater levels of perseverance and resilience among language learners, enabling them to surmount challenges and sustain motivation throughout the entire learning journey [30].

In a similar vein to the role of SSE in the relationship between CI and WTC, reciprocal determinism in SCT also contributes to the hypothesis that FLE mediates the influence of CI on WTC, as both SSE and FLE belong to personal beliefs (cognitive or emotional). Numerous existing studies [31–35] have examined the mediating role of FLE in EFL learning, such as its mediation between grit and foreign language performance, between trait emotional intelligence and perceived achievement, and between teacher enthusiasm and student engagement. Among prior research, the mediating role of FLE between classroom factors, such as classroom environment, class social climate, teacher support, and WTC is also frequently discussed. For instance, the study [16] examined the mediating role of emotions experienced in Chinese university students between classroom environment and WTC, finding that enjoyment has the largest mediating effect. Additionally, both studies [18,36] confirmed that FLE mediates the relationship between teacher-related factors, such as teacher support and teacher enthusiasm, and WTC.

The existing literature enhances our understanding of the role of FLE in mediating the relationship between classroom factors and WTC. However, knowledge remains limited, as the impact of FLE on the relationship between CI and WTC has yet to be fully explored. Despite the close relationship between concepts such as classroom environment, class social climate, and teacher support, they differ in nature from the concept of CI. Therefore, this study aims to address this gap and proposes the following hypothesis:

Hypothesis 3: Foreign language enjoyment functions as a mediating variable in the relationship between classroom interaction and willingness to communicate.

## 1.4 The relationship between classroom interaction, speaking self-efficacy, foreign language enjoyment, and willingness to communicate

Hypotheses 1, 2, and 3 suggest that CI holds a crucial position in influencing language learner's WTC, as CI furnishes an advantageous atmosphere for fostering both cognitive and emotional factors that are essential for communication. To be specific, CI exerts an influence on WTC through its effects on personal beliefs including SSE and FLE. Based on these hypotheses, it is reasonable to infer that if the influence of SSE on the FLE is confirmed, the chain mediation pathway of "CI-SSE-FLE-WTC" is valid. Relevant empirical studies also contribute to the understanding of the relationship between SSE and FLE. The study [37] identified a strong positive relationship between task-specific self-efficacy and enjoyment in EFL, although their correlation shifts across various time points. To further illustrate this relationship, research [38] validated that coping self-efficacy can positively affect FLE in the EFL context, thereby providing additional support for the potential link between SSE and FLE. Based on this evidence, the following hypothesis is proposed:

Hypothesis 4: Classroom interaction affects willingness to communicate through the chain mediating effect of speaking self-efficacy and foreign language enjoyment.

Building on these theoretical insights and the findings of previous empirical studies, the present study introduces a mediation model (Fig 1) to examine the relationship among classroom interaction, speaking self-efficacy, foreign language enjoyment, and willingness to communicate. This model seeks to provide empirical evidence and theoretical justification for the argument that improved classroom interaction can positively increase willingness to communicate through the individual mediating effects of speaking self-efficacy and foreign language enjoyment, as well as their chain mediating effect.

   

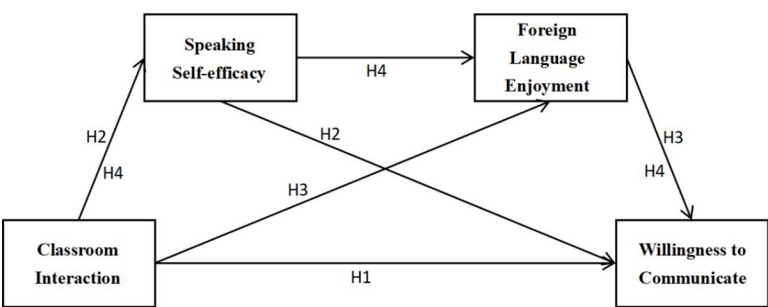

**Fig 1. Mediation model.**

## 2 Methods

### 2.1 Participants

The study initially recruited 679 undergraduate students from three regional comprehensive universities in northern China. These institutions, offering diverse programs in humanities, social sciences, and natural sciences, are situated in second- or third-tier cities and are representative of local socioeconomic and educational conditions. After excluding invalid responses, a valid sample of 623 students was retained. As detailed in Table 1, 326 participants (52.3%) are male and 297 (47.7%) are female. In terms of grade, 364 participants (58.4%) were freshmen, 97 (15.6%) sophomores, 82 (13.2%) juniors, and 80 (12.8%) seniors. Regarding academic disciplines, 290 participants (46.5%) were majoring in science, 153 (24.6%) in social science, and 180 (28.9%) in the humanities. The demographic data ensures diversity across gender, grade, and academic disciplines, enhancing the validity of the findings.

### 2.2 Measurement

The measurement used in this study was a self-designed questionnaire, consisting of 44 items grouped into four dimensions: classroom interaction, willingness to communicate in English, EFL speaking self-efficacy, and foreign language enjoyment. To ensure participants fully understand the survey, all the scale items, originally written in English, were translated into Chinese. The Chinese version was reviewed and verified by a scholar with a PhD in translation to ensure precision and reliability of the items. Prior to distributing the questionnaire on a large scale, a pilot study involving three participants was carried out to identify any ambiguous phrasing in the items. In consideration of the feedback they offered, particular revisions were introduced into the questionnaire to achieve a high level of clarity and validity. The final

**Table 1. Demographic information of the participants (No. = 623).**

| Variables | Level | Frequency | Percentage |
|---|---|---|---|
| Gender | Male | 326 | 52.3% |
| | Female | 297 | 47.7% |
| Grade | Freshman | 364 | 58.4% |
| | Sophomore | 97 | 15.6% |
| | Junior | 82 | 13.2% |
| | Senior | 80 | 12.8% |
| Academic Disciplines | Sciences | 290 | 46.5% |
| | Social Sciences | 153 | 24.6% |
| | The Humanities | 180 | 28.9% |

version of the questionnaire is organized into two sections. The first section collects demographic information from the respondents, containing gender, grade, and academic disciplines. The second section includes 44 items (see S1 Appendix) designed to assess students' perceptions from four dimensions: classroom interaction (items 1–11), willingness to communicate in English (items 12–21), EFL speaking self-efficacy (items 22–35), and foreign language enjoyment (items 36–44). In addition, each item is scored on a five-point Likert scale, ranging from "strongly disagree" (1) to "strongly agree" (5).

**2.2.1 Classroom interaction scale.** The classroom interaction scale was adapted from the interaction scale proposed by Wu and Gao, incorporating two of its dimensions: learner-instructor interaction and learner-learner interaction [39]. While the original scale was designed for online interactions, this study re-contextualized it for use in an off-line classroom environment. There are 7 items in the learner-instructor interaction dimension (e.g., "I often state my opinions to the instructor during the lecture.") and 4 items in the learner-learner interaction dimension (e.g., "I often ask other students questions during the lecture."). The reliability of this modified scale was established with a Cronbach's alpha coefficient of 0.889. Furthermore, the structural validity was verified through confirmatory factor analysis (CFA), which produced satisfactory model fit indices: $\chi^2/df = 4.733$, RMSEA $= 0.077$, GFI $= 0.939$, CFI $= 0.951$, and TLI $= 0.944$.

**2.2.2 Willingness to communicate in English scale.** The Willingness to Communicate in English Scale was used as the instrument to assess participants' perceptions. It was developed by Peng and Woodrow in their study on willingness to communicate in English in Chinese EFL classrooms, making it highly appropriate for the context of this research [40]. The scale includes 10 items related to the willingness to use English for activities such as role-playing, giving a short self-introduction, and translating spoken utterances (e.g., "I am willing to translate a spoken utterance from Chinese into English in my group."). This scale's reliability was notably high, evidenced by a Cronbach's alpha coefficient of 0.921, and its structural validity was supported by CFA results indicating a favorable model fit: $\chi^2/df = 4.512$, RMSEA $= 0.075$, GFI $= 0.944$, CFI $= 0.964$, and TLI $= 0.953$.

**2.2.3 Speaking self-efficacy scale.** This study borrowed 14 items from the scale to measure participants' speaking self-efficacy by Wang and Sun, covering the four self-efficacy dimensions [22]. Items 1–5 assess the participants' linguistic self-efficacy (e.g., "When speaking English in the classroom, I can speak with grammatical accuracy."); items 6–8 evaluate their self-regulatory efficacy (e.g., "When speaking English in the classroom, I can think of my goals before speaking."); items 9–10 measure their delivery self-efficacy (e.g., "I am not stressed out when speaking English in the classroom."); items 11–14 gauge their performance self-efficacy (e.g., "I can do an excellent job on the assignments and tests in the speaking course."). This scale demonstrated exceptionally good reliability, with a Cronbach's alpha coefficient of 0.939, and its structural validity was evidenced by CFA results revealing a good model fit: $\chi^2/df = 4.177$, RMSEA $= 0.071$, GFI $= 0.935$, CFI $= 0.966$, and TLI $= 0.956$.

**2.2.4 Foreign language enjoyment scale.** The present study adopted 9 items from the short version of Foreign Language Enjoyment Scale, developed by Botes et al. [41] on the basis of the original version introduced by Dewaele and MacIntyre [30]. The scale consists of three sub-scales: teacher appreciation sub-scale (items 1–3, e.g., "The teacher is friendly."), personal enjoyment sub-scale (items 4–6, e.g., "I've learned interesting things."), and social enjoyment sub-scale (items 7–9, e.g., "We form a tight group."). The reliability of this scale was confirmed with a Cronbach's alpha coefficient of 0.898, indicating good internal consistency. Additionally, the structural validity was affirmed through CFA, which yielded strong model fit indices: $\chi^2/df = 4.607$, RMSEA $= 0.076$, GFI $= 0.954$, CFI $= 0.963$, and TLI $= 0.950$.

## 2.3 Data collection

The research data was collected using the web-based questionnaire platform Wenjuanxing (https://www.wjx.cn/), and the dataset is available in S1 Data under the section of Supporting Information. Before distributing the questionnaire to

the participants via website link, the current study received approval from the Ethics Committee of North China University of Science and Technology (approval number: 2025025). The data was collected using convenient sampling over a three-day period, from May 13–15, 2025. Therefore, the generalizability of the findings should be interpreted with caution. Acknowledging this limitation could enhance the study's transparency. Throughout the collection process, clear instructions were provided in Chinese on how to complete the survey and assured participants that their responses and identities would remain confidential. Additionally, the participants were assured that their responses would be used exclusively for research purposes, which encouraged them to complete the questionnaire based on their genuine learning experience and feelings. All participants provided informed consent before taking part in the study. Consent was obtained in written form via a digital informed consent page embedded at the beginning of the online questionnaire, which participants read and agreed to before proceeding. All the participants had the right to withdraw from this research at any time, even if they had previously signed the informed consent form.

## 2.4 Data analysis

Data analysis in this study was conducted by means of IBM SPSS Statistics 27.0 and IBM Amos 28.0, including the assessments of reliability and validity, and the evaluation of structural equation model. First, descriptive statistics and Pearson correlation analysis were conducted to examine the means, standard deviations, and interrelationships among CI, WTC, SSE, and FLE. Based on these preliminary analyses, structural equation modeling (SEM) and bootstrap analysis with 5000 resamples were conducted to test the four proposed hypotheses. Hypothesis 1 was tested through path analysis, examining the direct effect of CI on WTC. For Hypotheses 2–4, mediation analysis was conducted to assess the indirect effects of CI on WTC through SSE and FLE, both separately and sequentially. The significance of these mediating effects was evaluated using 95% bootstrap confidence intervals. When the intervals did not include zero, the mediation paths were considered statistically significant.

## 3 Results

### 3.1 Descriptive statistics and correlation analysis

Descriptive statistics (means and standard deviations) and Pearson correlations for the four variables of CI, WTC, SSE, and FLE are presented in Table 2. The five-point Likert scale used in this study means that a higher score indicates a stronger tendency to agree. All the mean scores are over 3 for each of the four variables, suggesting most respondents tend to agree with the items. These statistics provide a basis for further analysis of the relationships and effects among these variables.

The results of Pearson correlation analysis revealed significant correlations among all the four variables, and these correlations aligned with the expected direction. To be specific, CI was positively correlated with WTC ($r = 0.429$, $p < 0.01$), SSE ($r = 0.411$, $p < 0.01$), and FLE ($r = 0.483$, $p < 0.01$); WTC showed a positive correlation with both SSE ($r = 0.414$, $p < 0.01$) and FLE ($r = 0.499$, $p < 0.01$); SSE exhibited a positive correlation with FLE ($r = 0.466$, $p < 0.01$).

Table 2. Means (M), standard deviations (SD), and correlations among the four variables.

| Variables | M | SD | CI | WTC | SSE | FLE |
|---|---|---|---|---|---|---|
| CI | 3.37 | 0.648 | 1 | | | |
| WTC | 3.473 | 0.689 | 0.429** | 1 | | |
| SSE | 3.163 | 0.681 | 0.411** | 0.414** | 1 | |
| FLE | 3.903 | 0.625 | 0.483** | 0.499** | 0.466** | 1 |

The symbol ** means $p < 0.01$.

## 3.2 Structural equation model analysis

### 3.2.1 Model fit testing.
To verify the four research hypotheses of this study, a structural equation model (SEM) was constructed, designating CI as the independent variable, WTC as the dependent variable, and SSE and FLE as the mediators. The SEM demonstrates good model fit indices: $\chi^2/df = 2.888$, RMSEA = 0.055, GFI = 0.949, CFI = 0.966, and TLI = 0.958. These indices suggest that the hypothesized relationships in the SEM are consistent with the observed data.

### 3.2.2 Path analysis between variables.
As is shown in Table 3, the path from CI to WTC is significant ($p < 0.001$), with a standard estimate of 0.329, indicating that CI exerts a significant positive impact on WTC. Therefore, the research hypothesis H1 is corroborated by the data. The path from CI to SSE is statistically significant ($p < 0.001$), with a standard estimate of 0.519, suggesting that CI has a strong positive effect on SSE. The relationship between CI and FLE is also significant ($p < 0.001$), supported by the standard estimate of 0.507, highlighting a positive influence of CI on FLE. With a standard estimate of 0.203 and a $p$-value less than 0.001, the path from SSE to FLE shows a significant positive influence of SSE on FLE. With a standard estimate 0.144 and a $p$-value of 0.002, the path from SSE to WTC is statistically significant, indicating a positive relationship. With a standard estimate 0.249 and a $p$-value less than 0.001, the path from FLE to WTC is highly significant, suggesting a substantial positive influence of FLE on WTC.

### 3.2.3 Mediation effect testing.
The results from bootstrap test for the significance of both direct effect and indirect effect are presented in Table 4. The results indicate that SSE and FLE have significant mediating effects, with the overall mediating effect value being 0.309. To be specific, the mediating effects are generated through three mediating chains. The mediation estimate for the "CI-SSE-WTC" path is 0.102, with a 95% bootstrap confidence interval ranging from 0.037 to 0.179, which does not include zero. This result indicates that SSE plays a significant role in the relationship between CI and WTC. Thus, the research hypothesis H2 is validated. The mediation estimate for the "CI-FLE-WTC" path is 0.171, with a 95% bootstrap confidence interval from 0.093 to 0.274, which excludes zero. This suggests that FLE significantly mediates the relationship between CI and WTC. Hence, the research hypothesis H3 is substantiated. With a mediation estimate of 0.036 and a 95% bootstrap confidence interval between 0.011 and 0.071, the results prove the significant chain mediating effect of SSE and FLE between CI and WTC. As a result, the research hypothesis H4 is confirmed.

**Table 3. Results of path analysis between variables.**

| Path | | | Estimate | Standard Error | Critical Ratio | $p$ | Standardized Estimate |
|---|---|---|---|---|---|---|---|
| WTC | <— | CI | 0.447 | 0.106 | 4.22 | *** | 0.329 |
| SSE | <— | CI | 0.656 | 0.076 | 8.618 | *** | 0.519 |
| FLE | <— | CI | 0.587 | 0.082 | 7.132 | *** | 0.507 |
| FLE | <— | SSE | 0.187 | 0.045 | 4.159 | *** | 0.203 |
| WTC | <— | SSE | 0.155 | 0.049 | 3.138 | 0.002 | 0.144 |
| WTC | <— | FLE | 0.291 | 0.063 | 4.603 | *** | 0.249 |

The symbol *** means $p < 0.001$.

**Table 4. Results of mediating effect analysis.**

| Path | Effect Types | Estimate | Lower | Upper | $p$ |
|---|---|---|---|---|---|
| CI=>WTC | direct effect | 0.447 | 0.21 | 0.765 | <0.001 |
| CI=>SSE=>WTC | indirect effect 1 | 0.102 | 0.037 | 0.179 | 0.003 |
| CI=>FLE=>WTC | indirect effect 2 | 0.171 | 0.093 | 0.274 | 0.001 |
| CI=>SSE=>FLE=>WTC | indirect effect 3 | 0.036 | 0.011 | 0.071 | 0.002 |
| CI=>WTC | total effect | 0.755 | 0.568 | 0.994 | <0.001 |

## 4 Discussion

### 4.1 Direct effect of CI on WTC

The present study confirms that CI exerts a significant positive impact on students' WTC in English. The findings provide empirical support for Hypothesis 1, which posits that increased classroom interaction fosters a higher willingness to communicate in English in EFL settings. This finding aligns with the predictions of social interactionist theory, which emphasizes the fundamental role of social interaction in language development [19]. Distinct from prior studies [6,11], which demonstrated that students' perceptions of CI influence their WTC, this study further advances the understanding of CI by shifting the focus from perceived classroom interaction (a cognitive variable) to actual classroom interaction (a behavioral variable). By doing so, it addresses a crucial gap in previous research and offers a more concrete understanding of how classroom engagement fosters WTC. Additionally, while previous studies [12,13] examined WTC across classroom contexts, their findings were limited by small sample sizes (8 or 23 participants), restricting their generalizability. In contrast, the present study, by employing a larger and more diverse sample (623 undergraduate students from three universities in China), extends previous findings and enhances the robustness of the relationship between CI and WTC. Thus, this research not only corroborates existing literature but also advances the field by refining the methodological approach and strengthening the empirical evidence on the role of CI in fostering WTC.

This finding contributes to the theoretical understanding of the relationship between CI and WTC by providing empirical validation of the social interactionist theory in an EFL context. While SIT emphasizes the importance of social interaction in cognitive and linguistic development [19], this study extends its applicability to WTC, demonstrating that CI not only facilitates language acquisition but also enhances students' readiness to engage in communication. From a pedagogical perspective, the finding underscores the necessity for educators to implement interaction-rich classroom environments that actively engage students in communicative activities. Given the significant role of CI in enhancing WTC, teachers should design instructional strategies that incorporate peer collaboration and teacher-student interactions to create an encouraging and supportive communicative atmosphere.

The direct effect of CI on WTC (standardized estimate = 0.329) accounts for approximately 43% of the total effect (0.755), indicating that while direct interaction is crucial, cognitive and emotional mediators (SSE/FLE) collectively explain a larger proportion (57%) of the relationship. This highlights the importance of addressing both behavioral engagement and psychological factors in fostering WTC. Building on this, the following three sections will examine the mediating roles of SSE and FLE in the relationship between CI and WTC, as well as explore how SSE and FLE jointly function as a chain mediator linking CI to WTC.

### 4.2 Mediating role of SSE between CI and WTC

SSE is found to significantly mediate the relationship between CI and students' WTC in English, as revealed by the present study. This finding provides empirical support for Hypothesis 2, which posits that SSE serves as a mediator linking CI to WTC. This study aligns with social cognitive theory, particularly Bandura's concept of reciprocal determinism [23], which views behavior as shaped by the interaction of personal factors, environmental contexts, and actions. Within this framework, CI acts as an environmental stimulus that strengthens students' SSE, which in turn promotes their WTC. This study advances existing research by specifically examining the mediating role of SSE in the relationship between CI and students' WTC, a link that has received limited attention in prior work. While previous studies [24–28] have demonstrated that self-efficacy mediates relationships between various aspects of the learning environment and outcomes such as academic engagement and achievement, they have primarily addressed general self-efficacy rather than context-specific constructs like SSE. As a result, the role of SSE in shaping communicative behavior remains under-explored. By foregrounding SSE in the CI-WTC relationship within an EFL context, the present study fills this research gap and extends the theoretical application of self-efficacy to communication-oriented outcomes, offering more targeted insights into how classroom environments foster students' communicative willingness.

 

From a practical standpoint, this finding suggests that educators should not only facilitate classroom interaction but also actively work to enhance students' speaking self-efficacy. Pedagogical strategies such as providing constructive feedback, modeling effective communication, encouraging peer collaboration, and creating opportunities for successful speaking experiences can strengthen students' confidence in their speaking abilities. As SSE significantly influences learners' WTC, cultivating this sense of efficacy can lead to more meaningful participation and improved communicative competence in EFL classrooms. Therefore, promoting SSE serves as a practical instructional goal that bridges interactional practices and learner outcomes in language education.

### 4.3 Mediating role of FLE between CI and WTC

The present study reveals that FLE significantly mediates the relationship between CI and students' WTC in English. This finding lends empirical support to Hypothesis 3, which assumes that FLE functions as a mediating variable linking CI and WTC. This mediation effect highlights the emotional mechanism through which the classroom social environment influences learners' communicative willingness. This study builds on and extends previous research by specifically examining how FLE mediates the relationship between CI and students' WTC, a connection that remains insufficiently examined. While earlier studies [31–35] have confirmed that FLE mediates relationships between learner traits (e.g., grit or emotional intelligence) and language performance or engagement, these focused primarily on individual characteristics rather than interactive classroom dynamics. Other studies [16,18,36] have explored FLE as a mediator between broader classroom factors (such as environment, social climate, and teacher support) and WTC. However, few studies have directly investigated the specific role of CI, a dynamic and participatory component of the classroom, in shaping learners' affective experiences and communicative behaviors. By focusing on CI and its emotional pathway to WTC via FLE, this study addresses a key gap in the literature. It contributes to a deeper understanding of how peer and teacher-student interaction fosters enjoyment and, in turn, promotes students' willingness to communicate. This extends existing research on FLE and enriches theoretical insights into affective mediation in communication-oriented EFL contexts.

From a theoretical perspective, this study reinforces Bandura's social cognitive theory [23], particularly the concept of reciprocal determinism, by demonstrating how FLE, an emotional aspect of personal belief, emerges from classroom interaction and influences communicative behavior (WTC). It expands SCT's explanatory scope by elucidating how emotional constructs such as enjoyment mediate the impact of social contexts on communicative behavior in EFL settings. Practically, these findings underscore the importance of fostering a classroom environment that actively promotes interaction and cultivates positive emotional experiences. Teachers are encouraged to design engaging, supportive tasks that stimulate enjoyment, as enhancing FLE can boost students' willingness to communicate and support more effective language learning.

After discussing the respective mediating roles of SSE and FLE between CI and WTC, it is equally valuable to explain why the mediating effect of FLE (0.171) is higher than that of SSE (0.102). This difference may be attributed to the distinct nature of these two constructs. The stronger mediating role of FLE may stem from its immediacy in classroom interactions — enjoyment arises from moment-to-moment positive experiences (e.g., peer collaboration, teacher praise), which directly motivate communication. FLE, fostered by supportive peer collaboration and teacher immediacy, enhances learners' WTC by creating a positive and risk-taking classroom atmosphere [42,43]. Additionally, experience-sampling evidence indicates that fluctuations in FLE are directly tied to instantaneous increases in communicative engagement [44]. Such emotional engagement can lead to greater willingness to take communicative risks and participate actively in class. In contrast, SSE, as a cognitive belief in one's communicative ability, tends to develop gradually through accumulated learning experiences and performance feedback, consistent with Bandura's model of self-efficacy [44]. As such, SSE may exert a more delayed or indirect influence on WTC.

### 4.4 Chain mediating role of SSE and FLE between CI and WTC

This study confirms the significant chain mediating effect of SSE and FLE in the relationship between CI and WTC. This suggests that CI not only directly influences WTC but also exerts an indirect effect through a sequential pathway, first

enhancing students' SSE, which in turn promotes FLE, ultimately increasing their WTC. This finding is consistent with theoretical assumptions grounded in social cognitive theory [23] and the social interactionist theory [19], which propose that environmental factors such as CI foster both cognitive (e.g., SSE) and emotional (e.g., FLE) components of personal belief systems essential for communication. While previous studies [16,31,34] have largely examined the individual mediating roles of either SSE or FLE, the combined and sequential influence of these two constructs has received little scholarly attention. Nonetheless, emerging empirical evidence has hinted at this potential link. The study [37] identified a positive relationship between task-specific self-efficacy and FLE, while the research [38] demonstrated that coping self-efficacy significantly predicts enjoyment in EFL contexts. Building upon this emerging literature, the present study empirically validates a chain mediation model where CI enhances WTC through the sequential development of SSE and FLE. This not only substantiates prior assumptions but also contributes a deeper understanding of how cognitive and emotional mediators interact to shape communicative willingness in second language learning.

Theoretically, the study extends SCT by illustrating how environmental input (CI) affects learner behavior (WTC) through interlinked personal factors: cognitive (SSE) and emotional (FLE). The findings offer a more integrative perspective on how self-efficacy and enjoyment function jointly within the communicative process, advancing theoretical models of affective and motivational development in EFL contexts. Practically, the study emphasizes the necessity of creating classroom environments that not only enhance students' speaking confidence but also foster enjoyment. Teachers should incorporate interactive tasks that provide opportunities for success and emotional engagement, thus nurturing both SSE and FLE. Such practices can strategically increase students' WTC, leading to more effective language learning and deeper classroom participation.

## 5 Limitations and future research directions

Despite the valuable findings of this study, several limitations should be acknowledged. First, the study was conducted exclusively in three universities in China, focusing on Chinese EFL learners within a specific socio-cultural and institutional context. As such, the findings may reflect context-bound patterns and should be interpreted cautiously when generalizing to other educational or cultural settings. Future research is encouraged to replicate and extend the current study across various educational institutions and socio-cultural contexts, both within and beyond China. Cross-national or cross-regional comparative studies could further examine whether the observed chain mediating effect of SSE and FLE between CI and WTC remain consistent across different learner populations, thereby enhancing the external validity of the findings.

Second, all variables in this study were measured using self-reported questionnaires, which may introduce potential biases such as social desirability or response bias. Participants might have provided answers they believed to be favorable or expected, rather than reflecting their actual experiences or beliefs. To address this limitation, future studies could adopt a multi-method approach by incorporating teacher evaluations, peer assessments, classroom observations, or language performance tasks alongside self-reports to triangulate the data.

Third, while this study applies Bandura's concept of reciprocal determinism to explore the mediating roles of SSE and FLE in the relationship between CI and WTC, it primarily focuses on a unidirectional pathway (CI-SSE/FLE-WTC). This approach only partially reflects the reciprocal nature of determinism, which emphasizes dynamic, bidirectional influences among personal, behavioral, and environmental factors. For instance, how students' WTC may, in turn, influence their SSE or FLE and reshape CI remains under-explored. Future research would benefit from exploring these reciprocal pathways and better capture the complex, interactive mechanisms proposed in social cognitive theory.

## Supporting information

**S1 Appendix.**   The complete 44 items in the questionnaire.
(DOCX)

**S1 Data.** The collected data from the questionnaire.
(XLSX)

## Author contributions

**Conceptualization:** Zhiyu Liu, Yanchao Yang.

**Data curation:** Lei Zhang, Xianzhi Wang.

**Formal analysis:** Zhiyu Liu, Lei Zhang.

**Methodology:** Zhiyu Liu, Yanchao Yang.

**Project administration:** Zhiyu Liu, Xianzhi Wang.

**Software:** Zhiyu Liu.

**Supervision:** Xianzhi Wang, Yanchao Yang.

**Validation:** Dawei Sun, Xianzhi Wang.

**Visualization:** Lei Zhang.

**Writing – original draft:** Zhiyu Liu.

**Writing – review & editing:** Dawei Sun, Yanchao Yang.

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
