## [Decision Letter · Decision Letter 0]

Dear Dr. Yang,

Thank you for submitting your manuscript to PLOS ONE. After careful consideration, we feel that it has merit but does not fully meet PLOS ONE’s publication criteria as it currently stands. Therefore, we invite you to submit a revised version of the manuscript that addresses the points raised during the review process.

We look forward to receiving your revised manuscript.

Kind regards,

Sijia Xue

Academic Editor

PLOS ONE

Journal Requirements:

“This research was funded by The Education Department of Hebei Province, China (grant numbers: 2024WYJG016 and 2024YYJG093)”

Reviewers' comments:

Reviewer's Responses to Questions

**Comments to the Author**

1. Is the manuscript technically sound, and do the data support the conclusions?

Reviewer #1: Yes

Reviewer #2: Yes

2. Has the statistical analysis been performed appropriately and rigorously?

Reviewer #1: Yes

Reviewer #2: Yes

3. Have the authors made all data underlying the findings in their manuscript fully available?

Reviewer #1: Yes

Reviewer #2: Yes

4. Is the manuscript presented in an intelligible fashion and written in standard English?

Reviewer #1: Yes

Reviewer #2: No

Reviewer #1: Thanks for presenting this well-written manuscript to the readers of Plos One. I enjoyed reading this paper. From reading this paper, I can tell the data were carefully analyzed and the findings are nicely presented. I only saw tiny issues that I would like to share with the authors.

1. In the abstract, I noticed some expressions that can be modified to further bring forth clarity. In the first sentence, why "while" is used to connect the two clauses. I cannot see a "while" relationship between them. When evaluating previous research, I recommend toning down a bit. Personally, I would not use the word limitations bluntly, but I also understand that different people have different writing styles. The authors can of course adhere to the way most comfortable to them. It can be nicer to say the limitations and directions explicitly. I am not saying that you should elaborate on them. A few phrases could add substances to the claim you make.

2. "In addition, studies (Bandura, 1997; Pekrun, 2006; Li et al., 2022; Hoesny et al., 2023; Yang et al.,

2024) motivate further...": I would like to know how these studies motivated further investigations.

3. I am not familiar with the in-house reference style of Plos One. It can be helpful to add doi to your references.

Reviewer #2: The manuscript addresses a timely and highly relevant topic within the field of second language acquisition (SLA), focusing on the behavioural influence of classroom interaction (CI) on students' willingness to communicate (WTC), with the mediating roles of speaking self-efficacy (SSE) and foreign language enjoyment (FLE). The topic is well-grounded in theoretical frameworks (Social Interactionist Theory and Social Cognitive Theory), and the study contributes empirical evidence to an area that is still developing. The writing is generally clear, the structure is logical, and the statistical analyses are appropriate for the proposed model.

Overall, the study is well-designed and offers meaningful contributions. However, there are several aspects where further clarification and minor revisions will significantly strengthen the manuscript, particularly in terms of methodological transparency, depth of discussion, and theoretical interpretation.

Given the solid foundation of the work and the minor nature of the revisions required, I recommend the manuscript for Minor Revision.

My full report is attached for your reference.

**Do you want your identity to be public for this peer review?** For information about this choice, including consent withdrawal, please see our Privacy Policy

Reviewer #1: No

Reviewer #2: No

---

## [Author Response · Author response to Decision Letter 1]

26 Jun 2025

Dear Editor,

Thanks for the thoughtful and constructive feedback from you and the reviewers. We sincerely appreciate the time and effort invested in evaluating our manuscript. The comments were invaluable in helping us refine and strengthen our work. In response, we have carefully revised the manuscript to address each requirement and comment, aiming to improve its clarity, coherence, and overall quality. A summary of the specific revisions is provided as below, including three parts: 1. Response to the journal requirements; 2. Response to Reviewer 1’s Comments; 3. Response to Reviewer 2’s Comments.

1. Response to the Journal Requirements

(1) Requirement 1: Please ensure that your manuscript meets PLOS ONE's style requirements, including those for file naming.

Response: The whole manuscript has been adjusted according to PLOS ONE's style requirements from the two recommended templates.

(2) Requirement 2: Thank you for stating the following financial disclosure:

“This research was funded by The Education Department of Hebei Province, China (grant numbers: 2024WYJG016 and 2024YYJG093)”

Response: The funder had no role in the study beyond providing financial support. The statement (The funder had no role in study design, data collection and analysis, decision to publish, or preparation of the manuscript.) has been added in the section of Funding.

(3) Requirement 3: Please include captions for your Supporting Information files at the end of your manuscript, and update any in-text citations to match accordingly. Please see our Supporting Information guidelines for more information: http://journals.plos.org/plosone/s/supporting-information.

Response: Captions for all Supporting Information files have been included at the end of the manuscript. These materials were appropriately cited within the manuscript text, with in-text citations accurately matched to their corresponding files.

(4) Requirement 4: Please review your reference list to ensure that it is complete and correct. If you have cited papers that have been retracted, please include the rationale for doing so in the manuscript text, or remove these references and replace them with relevant current references. Any changes to the reference list should be mentioned in the rebuttal letter that accompanies your revised manuscript. If you need to cite a retracted article, indicate the article’s retracted status in the References list and also include a citation and full reference for the retraction notice.

Response: The reference list has been carefully checked for accuracy and completeness. All entries have been formatted in accordance with PLOS ONE guidelines. This study does not cite any retracted articles.

2. Response to Reviewer 1’s Comments:

(1) Comment 1. In the abstract, I noticed some expressions that can be modified to further bring forth clarity. In the first sentence, why "while" is used to connect the two clauses. I cannot see a "while" relationship between them. When evaluating previous research, I recommend toning down a bit. Personally, I would not use the word limitations bluntly, but I also understand that different people have different writing styles. The authors can of course adhere to the way most comfortable to them. It can be nicer to say the limitations and directions explicitly. I am not saying that you should elaborate on them. A few phrases could add substances to the claim you make.

Response: First, we have replaced “while” with “and” to clarify that the two elements—classroom interaction (CI) and willingness to communicate (WTC)—are both important aspects of second language acquisition, rather than being contrasted. Second, in response to the reviewer’s suggestion to tone down the evaluation of previous research, we revised the sentence by replacing the original phrasing with “have not fully addressed.” This adjustment softens the critique and presents a more balanced and objective tone, acknowledging prior work while highlighting the research gap regarding the mediating roles of SSE and FLE in the relationship between CI and WTC.

(2) Comment 2. "In addition, studies (Bandura, 1997; Pekrun, 2006; Li et al., 2022; Hoesny et al., 2023; Yang et al.,2024) motivate further...": I would like to know how these studies motivated further investigations.

Response: Thank you for your helpful suggestion. In response, we have revised the paragraph to clarify how the cited studies motivate further investigation into the mediating roles of cognitive and affective factors. Specifically, Bandura’s self-efficacy theory and the control-value theory of achievement emotions provide key theoretical frameworks that highlight the importance of learners’ beliefs and emotions in language learning. Additionally, empirical studies have shown that classroom factors (e.g., teacher support, classroom environment) influence WTC through mediators such as SSE and FLE. These theoretical and empirical findings together justify the current study’s focus on the mediating roles of SSE and FLE in the CI-WTC relationship. Specifically refer to the revised part in the manuscript in Section Introduction, highlighted in green.

(3) Comment 3. I am not familiar with the in-house reference style of Plos One. It can be helpful to add doi to your references.

Response: We have added DOIs to references that are currently available only in their online versions.

3. Response to Reviewer 2’s Comments:

(1) Comment 1. Method -- Sampling and Participants

The manuscript reports that participants were drawn from three universities in China but does not specify:

The types of these institutions (e.g., comprehensive universities, research-intensive, teaching-oriented, regional vs national).

The geographical distribution of these institutions.

The sampling method used (e.g., convenience sampling, random sampling, stratified sampling).

Response: The types and regional characteristics of the sampled universities, and the sampling method were described in 2.1 Participants and in 2.3 Data Collection, respectively.

In 2.1 Participants:

The study initially recruited 679 undergraduate students from three regional comprehensive universities in northern China. These institutions, offering diverse programs in humanities, social sciences, and natural sciences, are situated in second- or third-tier cities and are representative of local socioeconomic and educational conditions.

In 2.3 Data Collection:

The data was collected using convenient sampling over a three-day period, from May 13 to 15, 2025. Therefore, the generalizability of the findings should be interpreted with caution. Acknowledging this limitation could enhance the study’s transparency.

(2) Comment 2. Questionnaire Development and Reproducibility

The questionnaire was largely self-developed or adapted but the full item list is not included.

Although the psychometric properties (reliability and validity) are reported and appear satisfactory, the absence of the full questionnaire limits the replicability of the study and prevents full scrutiny of item content.

Response: All 44 items in the questionnaire were provided in the section of Supporting Information.

(3) Comment 3. Discussion-2.1 Direct Effect vs. Mediating Effects Analysis (Section 4.1)

While the direct effect of CI on WTC is well-supported, the discussion would benefit from a clearer comparative analysis of the magnitude of direct vs. mediated pathways.

The total indirect effect (0.102 + 0.171 + 0.036 = 0.309) represents a substantial portion of the total effect (total effect = 0.755), yet this proportion is not directly highlighted.

Response: Comparative analysis of the strengths of direct (0.447) and indirect effects (0.309) was provided in the end of the section 4.1. The added paragraph not only compares the two different effect but also functions as a transition into Sections 4.2–4.4.

(4) Comment 4. Discussion-2.2 Differentiating SSE and FLE Mediation Strengths (Sections 4.2 and 4.3)

It is notable that FLE demonstrates a stronger mediating effect (0.171) than SSE (0.102), yet the manuscript does not attempt to interpret this difference.

Theoretically, affective states such as enjoyment may exert a more immediate influence on communication behaviour compared to cognitive constructs like self-efficacy, particularly in anxiety-prone contexts such as foreign language learning.

Response: We have addressed the reviewer’s suggestion by adding a clear theoretical explanation interpreting the difference in mediating effects between FLE and SSE. Please refer to the third paragraph in 4.3 Mediating role of FLE between CI and WTC.

(5) Comment 5. Potential Reciprocal Effects

While the manuscript appropriately applies Bandura’s concept of reciprocal determinism, the current model is unidirectional.

Given that WTC could feasibly influence CI (e.g., more willing students might engage more in class, further enriching interaction), a brief reflection on potential bidirectional relationships would strengthen the theoretical sophistication of the paper.

Response: A paragraph has been added to the Limitations section to acknowledge the potential reciprocal effects, particularly how WTC may influence CI. Corresponding recommendations for future research have also been provided. Please refer to the final paragraph of Section 5: Limitations and Future Research Directions.

6�Minor points:

1) Language and proofreading

Response: Some minor typographical and grammatical errors were already corrected. A thorough proofreading was also conducted before this submission. The revised parts were highlighted in green.

2) Consistency of terminology

Response: A thorough proofreading was made to ensure consistent use of terms, especially the consistent use of “classroom interaction” instead of “classroom communication”.

3�Figure presentation

Response: Figure 1 was already included in the manuscript and submitted separately as an individual file.

Once again, we sincerely thank you and the reviewers for your insightful guidance. Your feedback has been crucial in shaping this revision, and we are grateful for the constructive review process. We look forward to your response and hope our revisions align with your expectations.

Best regards.

---

## [Editor Report · Decision Letter 1]

The Impact of Classroom Interaction on Willingness to Communicate: The Mediating Roles of Speaking Self-Efficacy and Foreign Language Enjoyment

PONE-D-25-28227R1

Dear Dr. Yang,

We’re pleased to inform you that your manuscript has been judged scientifically suitable for publication and will be formally accepted for publication once it meets all outstanding technical requirements.

Kind regards,

Sijia Xue

Academic Editor

PLOS ONE
---

## [Editor Report · Acceptance letter]

PONE-D-25-28227R1

PLOS ONE

Dear Dr. Yang,

I'm pleased to inform you that your manuscript has been deemed suitable for publication in PLOS ONE. Congratulations! Your manuscript is now being handed over to our production team.

Kind regards,

on behalf of

Dr. Sijia Xue

Academic Editor

PLOS ONE